biochemistry/evolution/environmental science

marine turtles, collagen fingerprinting, species identification, zooarchaeology, ZooMS, aDNA

**Authors for correspondence:**
Virginia L. Harvey
e-mail: virginia.harvey@postgrad.manchester.ac.uk
Michael Buckley
e-mail: m.buckley@manchester.ac.uk

# Preserved collagen reveals species identity in archaeological marine turtle bones from Caribbean and Florida sites

Virginia L. Harvey[1], Michelle J. LeFebvre[2],
Susan D. deFrance[3], Casper Toftgaard[4,5],
Konstantina Drosou[6], Andrew C. Kitchener[7,8]
and Michael Buckley[1]

[1]Manchester Institute of Biotechnology, School of Earth and Environmental Sciences, University of Manchester, 131 Princess Street, Manchester M1 7DN, UK
[2]Florida Museum of Natural History, and [3]Department of Anthropology, University of Florida, Gainesville, FL 32611, USA
[4]The SAXO Institute, University of Copenhagen, Karen Blixens Plads 8, 2300 København S, Denmark
[5]National Museum of Denmark, Nationalmuseet, Ny Vestergade 10, Prinsens Palæ, DK-1471, København K, Denmark
[6]KNH Centre for Biomedical Egyptology, School of Biological Sciences, 99 Oxford Road, Manchester, M13 9PG, UK
[7]Department of Natural Sciences, National Museums Scotland, Chambers Street, Edinburgh EH1 1JF, UK
[8]The University of Edinburgh, Institute of Geography, School of Geosciences, Drummond Street, Edinburgh, EH8 9XP, UK

 VLH, 0000-0003-0796-8287; MJL, 0000-0002-1741-9997;
MB, 0000-0002-4166-8213

Advancements in molecular science are continually improving our knowledge of marine turtle biology and evolution. However, there are still considerable gaps in our understanding, such as past marine turtle distributions, which can benefit from advanced zooarchaeological analyses. Here, we apply collagen fingerprinting to 130 archaeological marine turtle bone samples up to approximately 2500 years old from the Caribbean and Florida's Gulf Coast for faunal identification, finding the vast majority of samples (88%) to contain preserved collagen despite deposition in the tropics. All samples can be identified to species-level with the exception of the Kemp's ridley (*Lepidochelys kempii*) and olive ridley (*L. olivacea*) turtles, which can be separated to genus level, having diverged from one another only approximately

5 Ma. Additionally, we identify a single homologous peptide that allows the separation of archaeological green turtle samples, *Chelonia* spp., into two distinct groups, which potentially signifies a difference in genetic stock. The majority of the archaeological samples are identified as green turtle (*Chelonia* spp.; 63%), with hawksbill (*Eretmochelys imbricata*; 17%) and ridley turtles (*Lepidochelys* spp.; 3%) making up smaller proportions of the assemblage. There were no molecular identifications of the loggerhead turtle (*Caretta caretta*) in the assemblage despite 9% of the samples being morphologically identified as such, highlighting the difficulties in relying on morphological identifications alone in archaeological remains. Finally, we present the first marine turtle molecular phylogeny using collagen (I) amino acid sequences and find our analyses match recent phylogenies based on nuclear and mitochondrial DNA. Our results highlight the advantage of using collagen fingerprinting to supplement morphological analyses of turtle bones and support the usefulness of this technique for assessing their past distributions across the Caribbean and Florida's Gulf Coast, especially in these tropical environments where DNA preservation may be poor.

# 1. Introduction

Marine turtles (Chelonioidea) have been swimming in our oceans for over 100 Myr [1]. Their fossil record first appeared in the early Cretaceous and modern forms of marine turtle have remained largely unaffected by the Cretaceous–Palaeogene extinction event—the fifth planetary mass extinction crisis [2]. Extant marine turtles have adapted to changing climate, glaciation and sea level throughout their evolution [1], and genetic analyses suggest these are likely to have resulted in periods of historical population dispersal, interspersed with restricted gene flow and subdivision (e.g. [3]). Globally, marine turtles have been exploited for millennia for their meat, eggs, shells and other products, and today they face further threats from habitat disturbances/modifications, poaching, pollution, rapid climate change and fisheries by-catch. Today, only seven species of marine turtle remain and six of these are classified as either vulnerable (olive ridley turtle, *Lepidochelys olivacea*; leatherback turtle, *Dermochelys coriacea*), endangered (green turtle, *Chelonia mydas*; loggerhead turtle, *Caretta caretta*) or critically endangered (Kemp's ridley turtle, *Lepidochelys kempii*; hawksbill turtle, *Eretmochelys imbricata*), with the seventh (flatback turtle, *Natator depressus*), as yet unclassified due to deficient data [4].

## 1.1. Marine turtle exploitation

Disturbance and degradation of habitats and breeding grounds have contributed to extensive marine turtle population declines globally (e.g. [5]). Marine turtles are highly migratory, long-lived and slow to mature (5–12 years in Kemp's ridleys [6]; and at least 22 years in loggerheads [7]), which makes them particularly vulnerable to anthropogenic pressures. Historic harvesting of marine turtles for local consumption is generally thought to have been sustainable (e.g. [8]), yet an increasing number of studies indicate that both prehistoric and historic-era subsistence and commercial marine turtle exploitation has had significant impacts on turtle populations, including the loss of entire nesting sites (e.g. [9]). For example, *C. mydas* and *E. imbricata* across the whole Caribbean were estimated in 2006 to have been reduced to a respective 0.33% and 0.27% of their historic abundance prior to European arrival in the region [5]. Today, both targeted [10] and accidental [11] removals of turtles from our oceans is conservatively estimated at several tens of thousands of individuals per year and removals at this scale are driving worldwide depletion in marine turtle abundance, with only relatively few local success stories (e.g. [8]).

## 1.2. Marine turtle phylogeography

Currently extant marine turtles (Chelonioidea) are classified into two families: Dermochelyidae with one extant species, the leatherback turtle (*D. coriacea*), and Cheloniidae with six; loggerhead (*C. caretta*), green (*C. mydas*), hawksbill (*E. imbricata*), Kemp's ridley (*L. kempii*), olive ridley (*L. olivacea*) and flatback (*N. depressus*) turtles. The flatback turtle has the smallest geographical range, with its distribution restricted to the continental shelf and coastal waters of Northern Australia, Southern Indonesia and Southern Papua New Guinea. Juveniles from all other marine turtle species exhibit an oceanic phase, which combined with strict thermal tolerances has driven ancient evolutionary partitioning between Indo-Pacific and Atlantic populations in the green and hawksbill turtles (figure 1; [17]). Conversely,

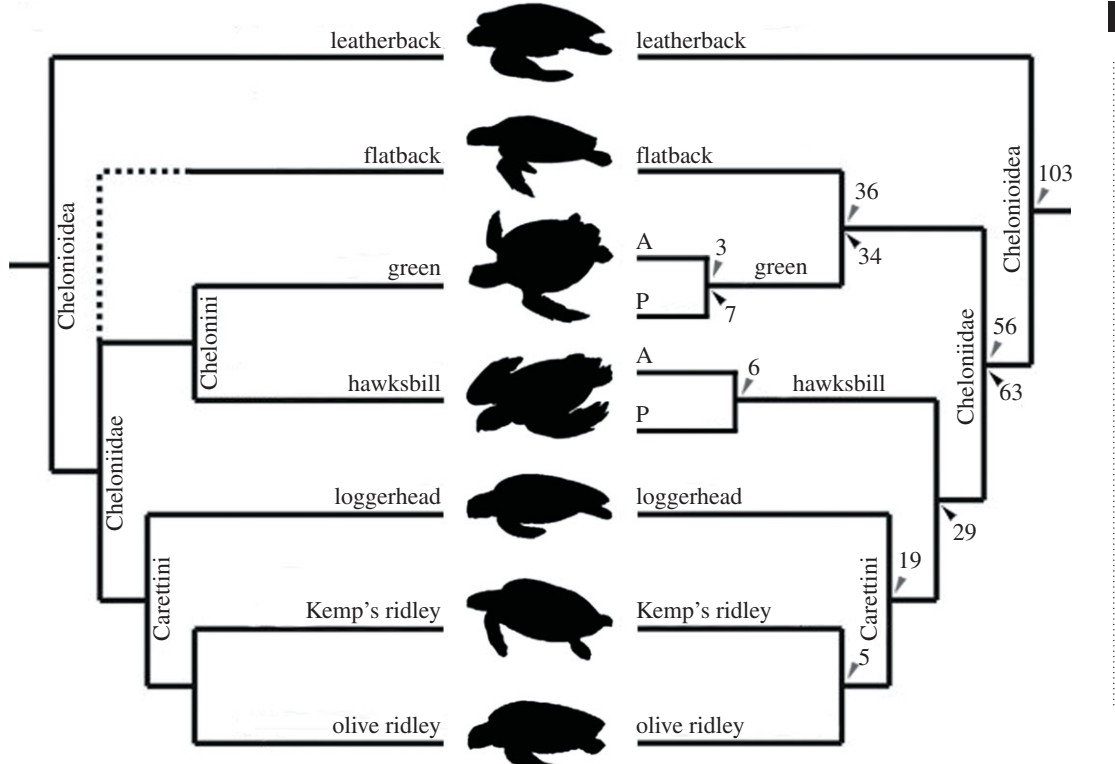

**Figure 1.** Schematic of the phylogenetic relationships between extant marine turtle species based upon (left) a morphological analysis by Gaffney & Meylan [12] (the flatback turtle was not included in the study) and (right) molecular analysis by Thomson & Shaffer [13]. Numbers represent estimated divergence times in millions of years (rounded to the nearest whole number), according to Naro-Maciel *et al.* [14] (black, bottom) and Duchene *et al.* [15] (grey, top). A, Atlantic clade; P, Pacific clade. Diagram adapted from Jones *et al.* [16]. Silhouettes and branch lengths not to scale.

molecular data show that the loggerhead and the two ridley species have undergone more recent inter-oceanic mixing and thus display less intraspecific genetic differentiation geographically [15]. Leatherbacks display a much greater thermal tolerance and are, therefore, less geographically restricted with no intraspecific partitioning of lineages, despite the fact that they separated from the other marine turtles over 100 Ma [15,17]. The leatherback turtle represents a more basal phylogenetic position according to both morphological and molecular analyses [14,18–20]. Molecular data support the divergence of two subfamilies within the Cheloniidae: the Chelonini (green and flatback turtles), and the Carettini (hawksbill, loggerhead and ridley turtles), splitting from each other 56–63 Ma [14,15], in a relationship reflected in more recent morphological studies [20], but not in earlier ones [13]. However, morphological and molecular analyses unanimously agree that the loggerhead and the ridley turtles form a monophyletic clade (e.g. [13,15]). Finally, earlier molecular phylogenetic studies support the grouping of the flatback turtle with the Carettini [18,19], but a more recent molecular consensus, including whole mtDNA genome sequencing, positions this species as sister to the green turtle [14,15].

Where there is now general agreement on the evolutionary relationships between the seven living marine turtle species, estimated divergence times between the lineages still remain relatively unresolved. In particular, the divergence timeframe between the Atlantic (A) and Pacific (P) lineages of green turtle (*C. mydas*) has been proposed by more recent molecular analyses to be 3.09 Ma (range of 1.76–4.87 Ma; [15]) and approximately 7 Ma (1.92–13.47 Ma; [14]) (figure 1), although notably with the latter study displaying a cautiously wide confidence range. As complete mitogenome analyses (as in [15]) have been shown to provide more reliable divergence timeframes than single mitochondrial markers (as in [14]), the more accurate divergence timeframe is likely to be around the 3 Ma estimate [15,21]. Interestingly, Duchene *et al.* [15] also report two distinct geographical lineages, Atlantic and Pacific, within the hawksbill turtles (*E. imbricata*) that are estimated to have diverged 5.63 Ma (3.44–8.85 Ma), an earlier separation time than that of the two disparate green turtle clades. However, Naro-Maciel *et al.* [14] noted no genetic lineage difference between the Pacific and Atlantic samples of hawksbill turtle they tested.

## 1.3. Historical baseline data for marine turtles

Establishing historical baselines is crucial for the understanding of long-term ecological changes and is becoming increasingly relevant in both conservation and modern management. Common issues associated with inaccurate, underestimated or 'shifting' baselines—when research uses modern population data as the baseline against which to measure change—include lower recovery targets, higher fishing quotas and more optimistic assessments of conservation status [22]. Previous marine historical data assessments have proved promising in addressing these issues by providing insight into extinction and extirpation patterns, population estimates, spatial shifts and species composition and size structure. Historical records suitable for understanding human impacts on marine species and ecosystems can include archival documents, early survey and monitoring records, interviews, zooarchaeological data and palaeontological data (see [22] for an overview), with the latter two providing perhaps the best opportunity for harnessing earlier data that may be thousands [23] or even millions of years old [24].

The three main methods of identifying ancient bones are morphological identification, DNA analysis and protein analysis, with proteomics becoming increasingly more popular in twenty-first century science due to the longevity of protein survival in the archaeological record [25]. At present, the morphological identification of bones is the most common method of specimen identification in zooarchaeology. Recent advancements in comparative marine turtle skeletal morphology have improved our ability to identify marine turtle carapace, plastron, cranial and limb elements recovered from archaeological assemblages [26]. However, in many archaeological contexts of deposition, adult and juvenile marine turtle remains are broken, highly fragmented, eroded and disassociated from the original location of capture or butchery. As a result, zooarchaeological marine turtle specimens may lack morphological markers diagnostic of particular species and/or have eroded surfaces and edges necessary to make confident genus- or species-level identifications. Moreover, artefacts shaped from marine turtle bones are often modified to the extent that it is not possible to morphologically identify the turtle species represented. Thus, morphological and biochemical-based methods for marine turtle bone identification provide complementary approaches to the advancement of zooarchaeological bone identification for obtaining species-level taxonomic identifications.

## 1.4. The role of collagen in species identification of ancient bones

Collagen type 1 ('collagen (I)'), a fibrous protein found ubiquitously in vertebrate bone, has been shown to survive longer post-deposition than many other informative biomolecules, including ancient DNA (aDNA) (e.g. [27], and see [28,29]). The high relative stability of the collagen protein through geological time and degree of amino acid sequence specificity between taxa has established collagen (I as indispensable in determining taxonomic identity from ancient bone remains that are at a higher risk of lacking both morphological characteristic (e.g. [23]) and high-quality aDNA [27]. Collagen (I) has also been shown to survive in ancient bone material from equatorial climates, including the Caribbean (e.g. up to approx. 1500 years old from the Cayman Islands; [30]).

Collagen fingerprinting (otherwise known as 'ZooMS' when applied to zooarchaeological bone material; [31]) is a method of extracting collagen (I), cleaving the protein into peptides with a tryptic enzyme and visualizing the peptides using soft-ionization mass spectrometry, amenable to the rapid analysis of thousands of samples (e.g. [32]). The resulting peptide mass spectrum, or 'collagen fingerprint', displays numerous peaks across the $x$-axis representing peptides of a particular mass-to-charge ratio ($m/z$), in turn, related to the amino acid sequence. Collagen (I) is a slowly evolving biomolecule whereby approximately one amino acid substitution occurs every 1–8 Myr depending upon the vertebrate class, with fish collagen (I) yielding the greatest variation, followed closely by herpetofauna [33]. To date, no subspecies have been shown to differ in their collagen fingerprints, including in fishes [23].

In this study, collagen fingerprinting was applied to archaeological marine turtle remains associated with human deposits from locations across the Caribbean and Florida's Gulf Coast. The aim of this work was to assess the extent to which collagen fingerprinting can taxonomically distinguish between the seven extant marine turtle species. We then aimed to determine whether collagen (I) analysis of archaeological bone samples can reveal information on the species harvested by ancient coastal inhabitants of the regions, and the geographical ranges of these species in the time period covered by the samples (approx. 500–2500 years ago). Our final objective was to test the phylogenetic relationships between the seven extant marine turtles using constructed collagen (I) amino acid sequences in order to compare this mode of obtaining phylogenetic information to previously published phylogenetic trees based on both morphological and molecular data.

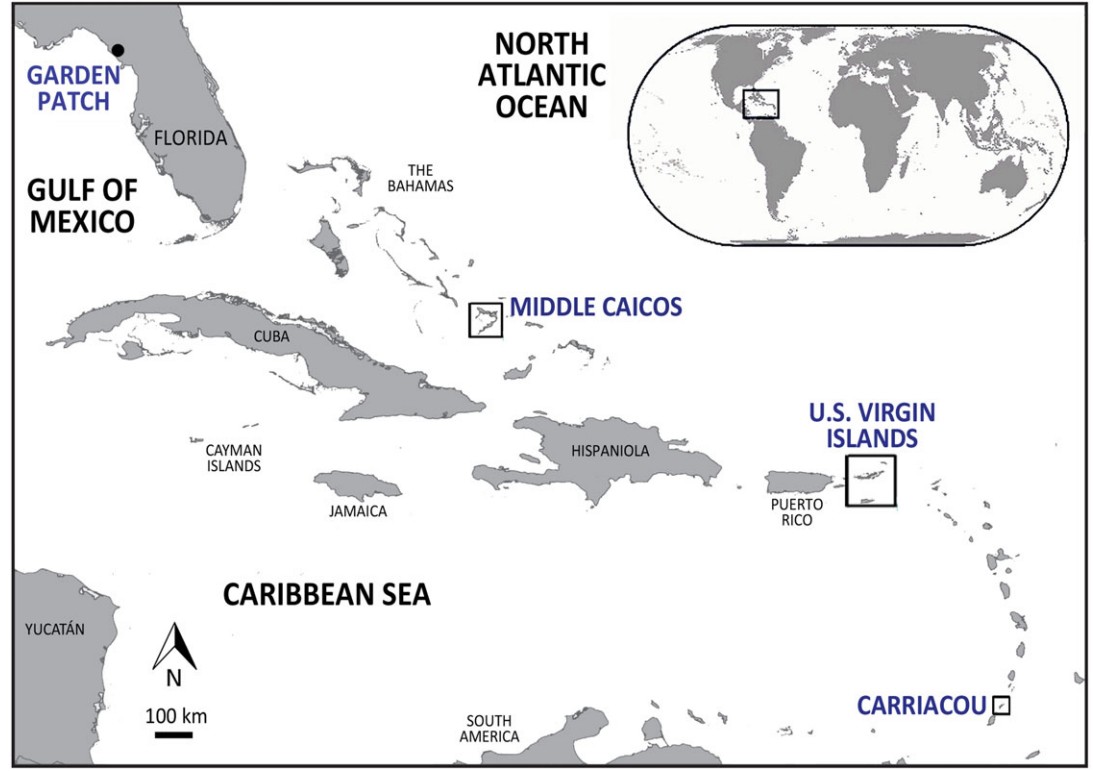

**Figure 2.** Map of the Caribbean and Florida's Gulf Coast showing broad geographical locations of the archaeological sites in this study (blue text), adapted from [34]. Inset: Location of the Caribbean (adapted from [35]).

# 2. Material and methods

## 2.1. Sampled archaeological sites

Archaeological bone samples were tested from seven sites chronologically spanning pre-Columbian (pre-AD 1492) and early Historic (post-AD 1492) time periods, including Garden Patch (Florida's Gulf Coast) and six sites in the Caribbean: Grand Bay (Carriacou), MC-6 (Middle Caicos), O.1 Magens Bay (St Thomas); and O.5 Coral Bay, O.7 Casey Long Bay and O.8 Little Cruz Bay (St John) (figure 2). The zooarchaeological analysis of marine turtle specimens from Garden Patch, Grand Bay and MC-6 was completed at the Florida Museum of Natural History (FLMNH), Gainesville. Zooarchaeological identifications were made through comparative morphological analysis using modern, complete, disarticulated marine turtle skeletons from the Environmental Archaeology Laboratory's comparative skeletal collection as well as marine turtle skeletons from the FLMNH herpetology collection. The archaeological marine turtle specimens from O.1 Magens Bay, O.5 Coral Bay, O.7 Casey Long Bay and O.8 Little Cruz Bay were not systematically analysed morphologically prior to this study. Identification of these specimens was to family level (Cheloniidae) and not beyond, based on conjectured identifications in the absence of morphological expertise. The assemblages consisted mainly of carapace and plastron fragments, with only a small representation of post-cranial and possible cranial fragments.

### 2.1.1. Garden Patch (Florida's Gulf Coast)

Garden Patch, situated near Horseshoe Beach on the west coast of Florida (USA), is a village-mound complex featuring seven mounds and extensive occupation areas [36,37]. The site was occupied from *ca* AD 100–1000 [38]. The Garden Patch faunal assemblage is taxonomically diverse with taxa from terrestrial, freshwater, estuarine and marine habitats. After fishes, turtles are the most abundant taxa present, including over 300 juvenile marine turtle elements recovered from both mound and village contexts [39]. Owing to morphological similarities, particularly at juvenile growth stages, 81% of the sampled marine turtle specimens from this site (50 of 62) have only been identified to family level

(Cheloniidae), with the remainder ($n = 12$) classified as cf. *C. caretta* (see electronic supplementary material, table S1).

### 2.1.2. Grand Bay (Carriacou)

Grand Bay is an archaeological village site characterized by extensive midden deposits, post-hole features, human burials and refuse pits. It is located on the southeastern coast of Carriacou (southern Grenadines) and was occupied during *ca* AD 400–1300 [40]. The midden is rich in marine taxa from inshore, coral reef and pelagic habitats [41], and the marine turtle assemblage from Grand Bay is remarkable in terms of the abundance and diversity of elemental representation, including carapace, plastron, cranial and limb elements. Net gauges shaped from marine turtle plastron have also been recovered from the site. Species previously identified from the Grand Bay zooarchaeological assemblage include green (*C. mydas*) and hawksbill (*E. imbricata*) turtles [42], both of which inhabit the reefs of Carriacou today [43]. Of the 11 Grand Bay bone samples in this study, 6 were morphologically identified to family level (Cheloniidae), and 5 to suspected species level (two *C. mydas* and three *E. imbricata*; electronic supplementary material, table S1).

### 2.1.3. O.1 Magens Bay (St Thomas), and O.5 Coral Bay, O.7 Casey Long Bay and O.8 Little Cruz Bay (St John; US Virgin Islands)

The zooarchaeological remains from St Thomas and St John originate from the first scientific archaeological expedition to the Virgin Islands in 1922 and 1923, and arguably to any Caribbean island [44]. Led by Dr Gudmund Hatt in 1922, the expedition excavated at more than 30 different sites, which allowed Hatt to establish a substantial research collection at the National Museum of Denmark (NMD) and propose the first archaeological chronology for the Caribbean Islands [45]. This study samples the two largest zooarchaeological assemblages from the collection; O.1 Magens Bay (St Thomas) and O.5 Coral Bay (St John), both midden sites with abundant ceramic and zooarchaeological deposits, alongside human burials. Two further sample sites, O.7 Casey Long Bay and O.8 Little Cruz Bay (St John), are both midden sites hosting smaller ceramic and zooarchaeological assemblages (C. Toftgaard 2016, unpublished data). Recent archaeological studies [46], in combination with radiocarbon and thermoluminescence dating of ceramic artefacts and organic refuse material from the four sites indicate occupation phases during: *ca* AD 500–1500 for O.1 Magens Bay; *ca* AD 600–1200 for O.5 Coral Bay; *ca* AD 500–900 for O.7 Casey Long Bay; and two separate occupation phases between *ca* 600 BC–AD 500 and *ca* AD 700–1100 for O.8 Little Cruz Bay [47]. The zooarchaeological assemblages were not used in Hatt's original research into the pre-Columbian Amerindian Cultures [45] and have lain dormant at the NMD ever since [46], with this study being the first to analyse a selection of the zooarchaeological remains from these sites. None of the samples from the four sites were morphologically identified beyond 'suspected Cheloniidae' prior to analysis.

### 2.1.4. MC-6 (Turks and Caicos)

Located in the centre of Middle Caicos in the Turks and Caicos Islands, MC-6 was occupied *ca* AD 1300–1650 and is composed of middens, a central plaza area, astronomical stone alignments, and features indicative of past structures [42,48]. This site also has evidence that it functioned as a specialized salt extraction locale [49]. The zooarchaeological assemblage from the site shows that faunal exploitation was focused on marine habitats, particularly the flats of the Caicos Bay [48]. Owing to fragmentation, marine turtle specimens from the site ($n = 4$) were not identified beyond Cheloniidae (see [49]).

## 2.2. Biomolecular analysis

### 2.2.1. Archaeological material

Collagen fingerprinting was achieved following the methods detailed in van der Sluis *et al.,* [50]. In brief, 130 archaeological bone samples (62 from Garden Patch, 11 from Grand Bay, 4 from MC-6, 35 from O.1 Magens Bay, 13 from O.5 Coral Bay, 2 from O.7 Casey Long Bay and 3 from O.8 Little Cruz Bay) were drilled using a sterilized drill bit per sample. Approximately 50 mg bone powder was collected and demineralized in 0.6 M hydrochloric acid (HCl) for 20 h before the supernatant was transferred into

100 µl of 50 mM ammonium bicarbonate (ABC) via 10 kDa molecular weight cut-off (MWCO) ultrafilters. Samples were then digested with 0.4 µg sequencing grade trypsin at 37°C for 18 h, before being diluted and co-crystallized with α-cyano-4-hydroxycinnamic acid matrix. Samples were analysed using a Bruker Ultraflex II MALDI mass spectrometer operating at up to 2000 laser shots per sample spot. Samples that generated poor spectra were purified into 10 and 50% acetonitrile (ACN) fractions, using C18 ZipTip pipette tips, and were air-dried before being resuspended in 10 µl 0.1% trifluoroacetic acid and analysed as above. One sample, GP9, identified as *Chelonia* sp., was submitted for LC-MS/MS analysis (Waters nanoAcquity UPLC system coupled to a Thermo Scientific Orbitrap Elite MS) to assist with peptide and sequence identification following Buckley *et al.* [51]. Additionally, six samples were selected for aDNA analysis (GP9, GP10, GP20, GP56 from Garden Patch, and GB64 and GB72 from Grand Bay). Samples GP10 and GB64 were identified through collagen fingerprinting as *C. mydas*, and samples GP9, GP20, GB64 and GB72 were identified as *Chelonia* sp., suspected as being from a different genetic stock (see electronic supplementary material, S1 for full aDNA methodology).

### 2.2.2. Modern reference material

To obtain reference collagen fingerprints, we tested at least one bone sample from all seven extant marine turtle species, *C. mydas* (*n* = 3; Vietnam, Pacific; Florida, Atlantic; Mexico, Pacific), *N. depressus* (*n* = 1; Australia), *C. caretta* (*n* = 1; Florida), *E. imbricata* (*n* = 1; Puerto Rico, Atlantic), *L. kempii* (*n* = 1; Florida), *L. olivacea* (*n* = 1; Costa Rica) and *D. coriacea* (*n* = 1; Florida) (electronic supplementary material, table S2). Additionally, 23 testudine bone samples from Chelydridae, Trionychidae, Emydidae, Testudinidae and Trionychidae families were acquired for analysis to assist in the identification of non-marine turtle samples (electronic supplementary material, table S2). Samples were analysed for their collagen fingerprints as above, fractioning into 10 and 50% ACN using C18 tips. Combined fractions from eight specimens (*C. mydas* [Atlantic], *C. mydas* [Pacific], *N. depressus*, *C. caretta*, *E. imbricata*, *L. kempii*, *L. olivacea*, *D. coriacea*) were submitted for LC-MS/MS analysis as above to assist with peptide sequencing. Collagen (I) amino acid sequences of α1 and α2 chains from *C. mydas* were obtained using online protein search tool BLASTP (Basic Local Alignment Search Tool). Error-tolerant (ET) and decoy searches were undertaken against this sequence, following Harvey *et al.* [52] and references therein, to obtain probability-matched sequence data for *N. depressus*, *C. caretta*, *E. imbricata*, *L. kempii*, *L. olivacea* and *D. coriacea*. ET searches used the following criteria: peptide tolerance of ±5 ppm, MS/MS fragment ion mass value tolerance of 0.5 Da, fixed carbamidomethyl modification of cysteine (mass shift = +57.02 Da), variable modifications for the oxidation of lysine (K) and proline (P) (mass shift = +15.99 Da) and the allowance of one missed cleavage. A final decoy search was used to mine the finished sequences, filtered to only include tryptic peptides that scored higher than the highest false-positive result with all other positions replaced with an 'X'. Applied decoy search criteria was as above except with two missed cleavages, additional variable deamidation of asparagine (N) and glutamine (Q) modifications (mass shift = +0.98 Da) and variable oxidation of methionine (M). Fragment ion spectra for novel amino acid substitution biomarkers were manually examined for quality before selection (electronic supplementary material, figure S2). Where the isobaric residues leucine (L) and isoleucine (I), provided ambiguity, sequences were completed with the residue L to provide consistency. We found no ambiguity between lysine (K) and glutamine (Q) residues as these sites are highly conserved in marine turtles, with the former also providing a tryptic digest site. Where the post-translational modification (PTM) of hydroxylation provided ambiguity with an alanine (A) to serine (S) substitution (mass shift of either = +15.99 Da), the sequence was completed with the residue congruent with the published sequence for *C. mydas* (BLASTP).

Phylogenetic analyses were performed with MEGA-X (v. 10.0.4), using the maximum-likelihood (ML) statistical method and mtREV + F model for substitution, as found by the software to be most appropriate for the dataset. A total of 10 000 bootstrap replications were carried out with gamma distribution (four categories), complete deletion (i.e. any amino acid missing for one species is replaced with an 'X' for all others), and with *Pelodiscus sinensis* as an outgroup.

# 3. Results and discussion

## 3.1. Taxonomic resolution of collagen fingerprinting in marine turtles

All seven modern marine turtle samples produced collagen fingerprints, and in-depth analysis reveals that each contains a combination of collagen (I) biomarkers that are unique to each genus (table 1;

**Table 1.** A selection of collagen (I) biomarkers from extant marine turtles, highlighting markers that are unique to a particular genus or species in bold (peptide code labels homologous to [53]).

| collagen (I) peptide code | Chelonia mydas | Natator depressus | Caretta caretta | Eretmochelys imbricate | Lepidochelys spp. | Dermochelys coriacea |
|---|---|---|---|---|---|---|
| A1T47 | 1136 | 1136 | 1136 | 1136 | **1152** | 1136 |
| A2T85 (A) | 1220 | 1220 | 1220 | 1220 | 1220 | 1220 |
| A1T28 | 1393 | 1393 | 1393 | 1393 | 1393 | 1393 |
| A2T43 (B) | 1453 | 1453 | 1453 | **1443** | 1453 | 1453 |
| A1T62 | 1459 | 1459 | 1459 | 1459 | 1459 | 1459 |
| A1T79 | **1516** | 1490 | 1490 | 1490 | 1490 | 1490 |
| A1T21 | 1572 | 1572 | 1572 | 1572 | 1572 | 1572 |
| A2T69 (D) | 2097 | **2107** | 2097 | 2097 | 2097 | **2125** |
| A2T65/66 | **2353** | 2341 | 2341 | 2341 | 2341 | **2311** |
| A2T60 | 2455 | 2455 | 2455 | **2485** | 2455 | **2510** |
| A1T85 | 2705 | 2705 | 2705 | 2705 | 2705 | 2705 |
| A2T41/42 | 2790 | 2790 | 2790 | 2790 | 2790 | 2790 |
| A1T55/56 (F) | 2843/59 | 2869/85 | 2869/85 | 2843/59 | 2843/59 | **2853/69** |
| A2T67 (G) | 2929 | 2929 | 2929 | 2929 | 2929 | **2899** |
| A2T3 | 3007 | 3007 | 3007 | 3007 | 3007 | 3007 |

electronic supplementary material, figures S1–S3). The leatherback turtle, *D. coriacea* contains more unique peptide biomarkers than any other species, which is unsurprising considering its divergence time of over 100 Ma from the Cheloniidae (table 1; electronic supplementary material, table S3). We cannot currently verify any amino acid substitutions that separate the two ridley turtles (*Lepidochelys* spp.), even though they diverged from one another approximately 5 Ma [15,18] and are thus within the predicted timeframe for amino acid substitutions occurring in reptiles (more than one amino acid substitution per million years between green lizard *Anolis* and Chinese softshell turtle *Pelodiscus*) [33]. The inability of collagen fingerprinting to separate them here suggests a much slower rate of testudine collagen evolution compared with other reptilian groups (at least lepidosaurs). It could also be possible that both evolutionary pressures and biological constraints, for example, long generation times, may affect collagen (I) sequence evolution in marine testudines, particularly when compared with terrestrial representatives of the same order.

## 3.2. Faunal composition of archaeological turtles

The vast majority (114 of 130; 88%) of the archaeological marine turtle samples generated collagen fingerprints—a proportion that was much higher than anticipated given (i) the proposed age of the bones at ≥500–2500 years old, and (ii) the tropical (high temperature and high humidity) region in which the bones were deposited, which does not typically favour molecular preservation [54]. Using the reference collagen fingerprints, LC-MS/MS data and subsequent biomarkers, most of the samples were identified as green turtle, *Chelonia* spp. (63%). The assemblages also contain hawksbill (*E. imbricata*; 17%) and ridley specimens (*Lepidochelys* spp.; 3%), with the latter only found at Garden Patch (table 2; electronic supplementary material, figures S1). The presence of these species in the assemblage was expected based on (i) the results from previous morphological studies (e.g. [42]), and (ii) considering that the geographical ranges of these species cover these same localities today. However, the absence of the loggerhead turtle, *C. caretta*, was not specifically predicted, particularly as a number of samples (*n* = 12; Garden Patch samples 1, 2, 8–17) were morphologically identified, albeit cautiously, as *C. caretta* prior to collagen analysis (electronic supplementary material, table S1). These misidentifications are unsurprising given that the archaeological marine turtle assemblage from Garden Patch is dominated by juvenile remains, which are notably difficult to identify morphologically [39]. The full assemblage also did not contain any leatherback turtle specimens, despite the localities falling well within the range of this species. However, the occurrence of

leatherbacks in archaeological collections is generally very rare, so far recorded only from Panama (AD 600–900; [55]) and from St Thomas (AD 300–700; [56]).

Interestingly, the assemblage also contains six samples of non-marine turtles, previously identified through morphology as Cheloniidae and identified here to family level using collagen fingerprinting: one Chelydridae (snapping turtles and relatives), two Testudinidae (tortoises) and three Emydidae (terrapins and relatives) (table 2; electronic supplementary material, table S1). The majority of these non-marine samples come from Garden Patch (four of six), with the remaining two coming from O.1 Magens Bay on St Thomas, from different excavation squares and depths. The remains of these species are likely to have been mistaken for juvenile marine turtles owing to their smaller size and deposition among true juvenile marine turtle species.

Curiously, of the 82 archaeological samples identified as green turtle, a single peptide (COL1A2T3), shows two different variations, appearing at $m/z$ 3007 (sequence GAPGTSGPPGAQGFQGPAGEPGEP GQTGPVGAR) in both the modern samples and the majority of the ancient samples ($n = 78$) and at $m/z$ 3035 (suspected sequence GVPGTSGPPGAQGFQGPAGEPGEPGQTGPVGAR) in a minority of archaeological samples from the Garden Patch ($n = 3$) and Grand Bay ($n = 1$) sites (table 2; figure 3; electronic supplementary material, figures 2.30–31). The exchange of an alanine (A) amino acid to a valine (V) is a ±28 shift, which can be visualized through the peak positions in the collagen fingerprint (figure 3, inset). Other protein PTMs that can cause a ±28 mass shift and would be relevant to the sequences above include arginine asymmetric (aDMA) and symmetric (sDMA) di-methylation (mass shift 28.0312 Da) [57]; however, LC-MS/MS analysis did not report this modification as present in our samples. Formylation is also another modification that could occur, particularly on the N-terminus, where further sequence clarification of this peptide is needed to rule this out. There are no other peptide peaks that appear to have shifted in this manner between the two variants of Chelonia spp. Moreover, all three modern specimens of C. mydas, encompassing both Atlantic (UF42972) and Pacific (FLMNH 57247; FLMNH 57247) lineages, display the COL1A2T3 peak at $m/z$ 3007, identical to the vast majority of the archaeological samples. Therefore, we speculate that the samples with the $m/z$ 3035 visible peak, in place of $m/z$ 3007, may be from a different genetic stock. Although our attempts at amplifying aDNA were not successful (electronic supplementary material, S1), presumably owing to the age and origin of the bones, we cannot rule out the possibility that these samples may represent an extinct and/or undescribed species of marine turtle that is most closely related to C. mydas. LC-MS/MS analyses of both the Atlantic and Pacific clades of modern C. mydas do not highlight any differences in the collagen (I) sequences, and considering the two diverged a minimum of approximately 3 Ma [15], we suggest that the archaeological Chelonia sp., if indeed a distinct lineage, is likely to have diverged from C. mydas before this date.

Specimens of ridley turtles (Lepidochelys spp.) are only present at the Garden Patch site with a total of four elements. This is representative of their nesting habitat, which lies along the western edge of the Gulf of Mexico, along the Florida coastline, and northwards along the coasts of Georgia and South Carolina, USA [58]. By contrast, the hawksbill, E. imbricata, is not present at this site, but is present at all others excluding MC-6 (possibly due to a low sample size), corresponding to its nesting range from southernmost Florida (e.g. Dry Tortugas National Park; [59]), and across the wider Caribbean (figure 4).

Finally, from the archaeological bone samples that did not generate a collagen fingerprint, all 16 were morphologically identified to a family level (Cheloniidae), of which three were further refined to species—one as C. mydas and two as cf. C. caretta, although the loggerhead identifications should be looked upon with caution considering other samples morphologically classified as such were subsequently identified as C. mydas using collagen fingerprinting.

## 3.3. A molecular phylogeny of marine turtles using collagen proteomic sequencing

Phylogenetic reconstruction achieved here using collagen (I) amino acid sequences substantiates the findings of more recent nuclear and mitochondrial analyses by Naro-Maciel et al. [14] and Duchene et al. [15] (figure 5). In particular, our analyses support the placement of the flatback turtle, N. depressus, as sister to the green turtle, C. mydas and thus within the Chelonini tribe, rather than within the Carettini tribe as in more dated studies such as Dutton et al. [19] and Bowen et al. [18]. Our phylogeny also supports the distinct grouping of the Chelonini (green and flatback turtles) and Carettini tribes (hawksbill, loggerhead and ridley turtles), supported previously by both molecular [14,15] and morphological studies [20]. Finally, our study reinforces the close affiliation between the loggerhead and ridleys turtles, as well as the position of the leatherback turtle as the most basal of the marine turtles [13,15,20]. There were no differences between the collagen sequences of the

**Table 2.** The number (NISP) of collagen-derived Testudines identifications for each site featured in this study. See text for details on *C. mydas* and *Chelonia* sp.

| | Chelonia mydas | Chelonia sp. | Eretmochelys imbricata | Lepidochelys spp. | Chelydridae | Testudinidae | Emydidae | poor | total |
|---|---|---|---|---|---|---|---|---|---|
| Garden Patch, Florida | 42 | 3 | 0 | 4 | 1 | 2 | 1 | 9 | 62 |
| Grand Bay, Carriacou | 5 | 1 | 4 | 0 | 0 | 0 | 0 | 1 | 11 |
| MC-6, Turks and Caicos | 3 | 0 | 0 | 0 | 0 | 0 | 0 | 1 | 4 |
| 0.1 Magens Bay, St Thomas | 18 | 0 | 14 | 0 | 0 | 0 | 2 | 1 | 35 |
| 0.5 Coral Bay, St John | 7 | 0 | 2 | 0 | 0 | 0 | 0 | 4 | 13 |
| 0.7 Casey Long Bay, St John | 0 | 0 | 2 | 0 | 0 | 0 | 0 | 0 | 2 |
| 0.8 Little Cruz Bay, St John | 3 | 0 | 0 | 0 | 0 | 0 | 0 | 0 | 3 |
| total | 78 | 4 | 22 | 4 | 1 | 2 | 3 | 16 | 130 |

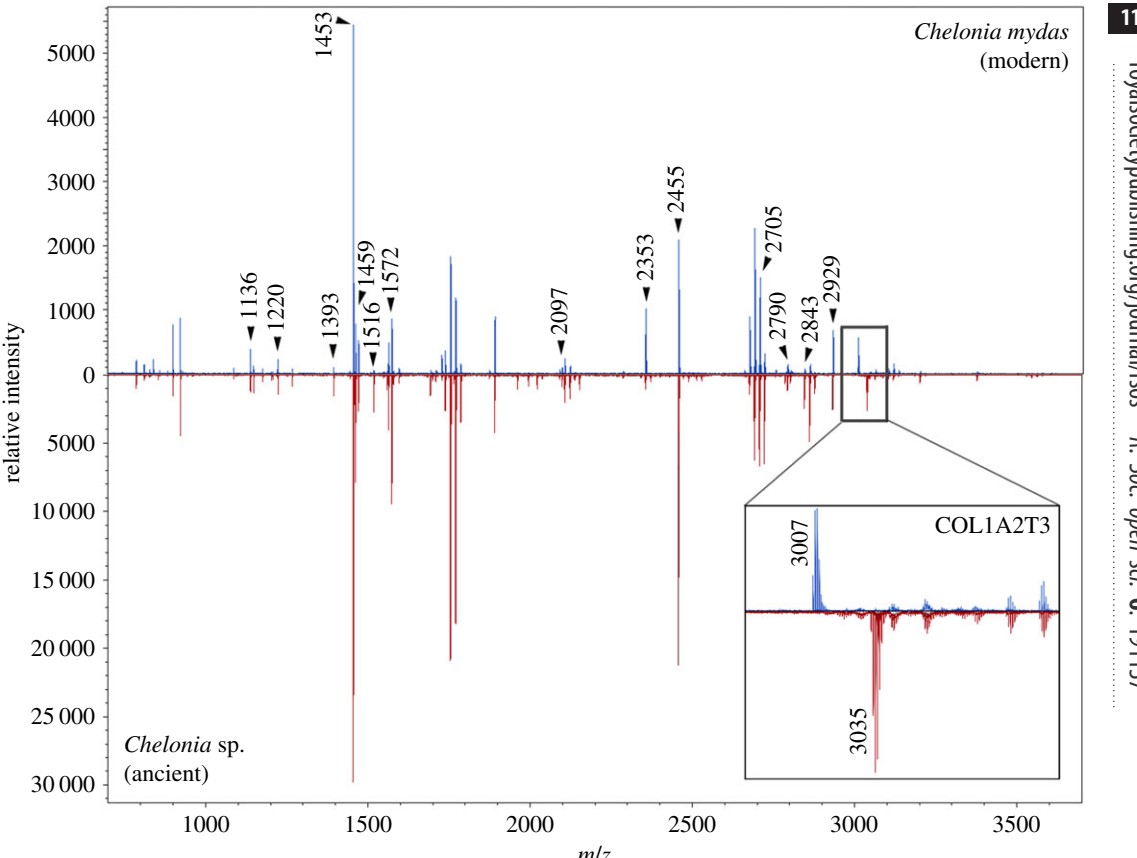

**Figure 3.** Collagen fingerprints (10 and 50% combined fractions) for the modern *C. mydas* sample (top; Pacific clade) and ancient sample 'GP9' of *Chelonia* sp., suspected to be from a different genetic stock (bottom), showing labelled biomarkers from table 1. Inset: expanded section of the spectrum showing the peak shift from *m/z* 3007 to *m/z* 3035 for the peptide COL1A2T3 (see text for sequences).

reference Pacific and Atlantic green turtles, and consequently these specimens form a clade with the full *C. mydas* sequence obtained from BLASTP. The variation of *Chelonia* sp., with one amino acid substitution, forms a monophyletic clade with the green turtle. Despite being based on a highly conservative dataset whereby the sequences were analysed through 'complete deletion' (any amino acid missing from one species is replaced with an 'X' for all others), the recovered topology matches well with previous publications using other sources of variation, lending further support to the use of proteomics-derived collagen sequence data for phylogenetic reconstruction [60,61].

## 3.4. The archaeological importance of turtle remains from coastal and island sites

The archaeological marine turtle samples in this study represent the species harvested at these localities by various pre-Columbian indigenous peoples from to approximately 600 BC to AD 1500. Generally, the large quantities of turtle remains signify their high importance as a dietary component and one that was potentially relied upon in these regions for two millennia. The taxonomic specificity of collagen (I) sequences, plus their longevity in ancient bone fragments, emphasizes collagen fingerprinting as a valuable tool for assessing long-term anthropogenic impacts and historical baseline estimates. Of particular importance, our study highlights the potential discovery of a different genetic lineage of marine turtle most closely related to the green turtle, *C. mydas*, which was harvested from the waters around the Florida coast and Carriacou at the time of human occupation. Both MALDI-MS and LC-MS/MS analyses on three modern green turtle specimens (covering Atlantic and Pacific variants), plus four archaeological *Chelonia* sp. specimens from two different sites support this claim. Although our attempts to amplify aDNA have been unsuccessful, the fact that this may represent an undescribed species of extinct marine turtle should not be overlooked.

At all sites, only around half of the marine turtle species available in each region appear to have been targeted by inhabitants. For example, at the site of Carriacou in the southern Caribbean, both green and

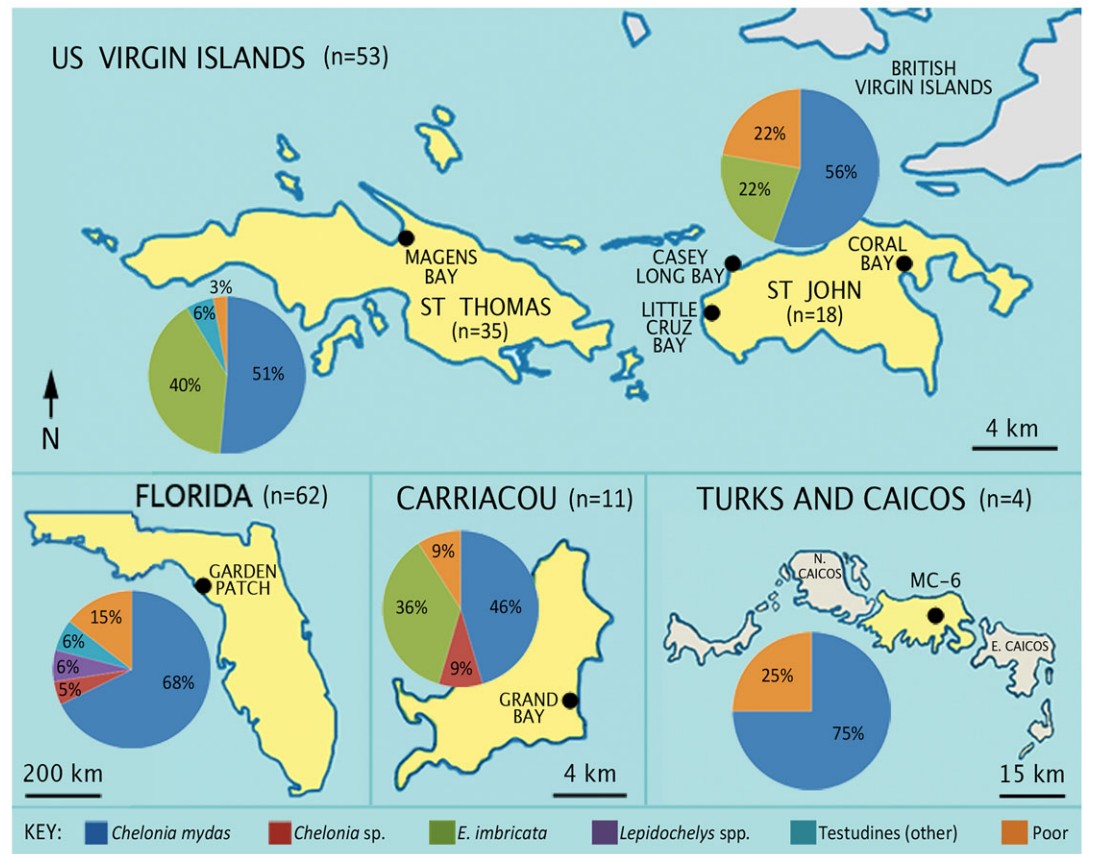

**Figure 4.** Individual maps of the archaeological sample sites in this study, including a pie chart representing identifications (NISP) as a percentage from each island.

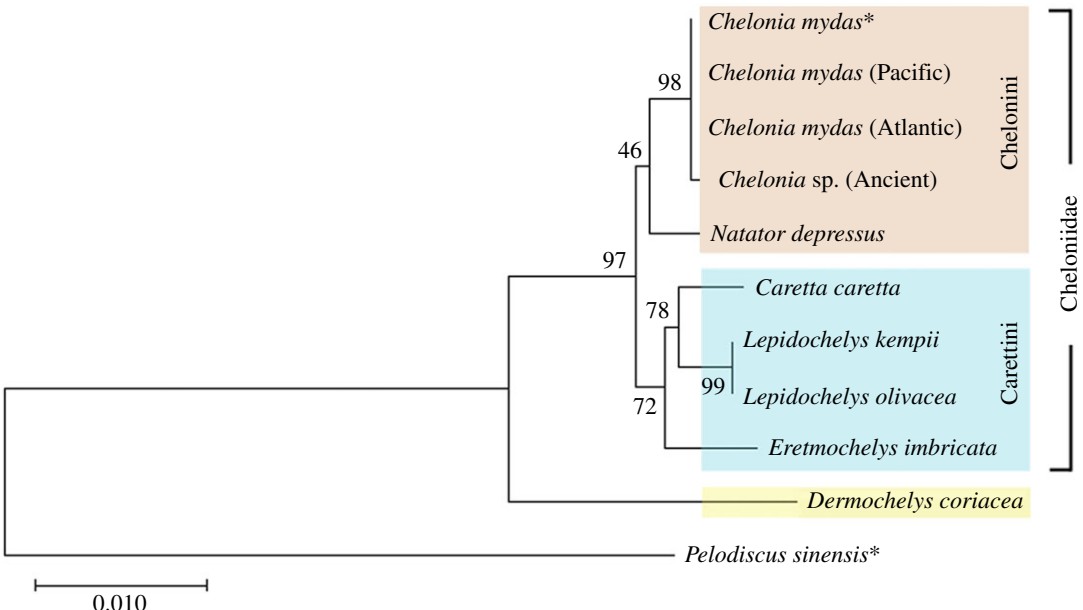

**Figure 5.** ML phylogenetic analysis of collagen (I) sequences from the order Testudines, including sequences extracted from BLASTP (*) and those obtained by LC-MS/MS during this study. Bootstrap values are shown next to each node.

hawksbill turtles were targeted, but we find no evidence of either leatherback or loggerhead turtles. These are the first data to indicate that specific turtle taxa were selected for capture in these islands. Based on analogy with modern marine turtle capture methods, archaeological turtles were probably hunted with a combination of nets in shallow seagrass beds and the use of spears on reef habitats

(e.g. [43]). Beach harvesting of nesting females may also have taken place (e.g. [41]), as well as the capture of juveniles, as suggested by the numerous juvenile bone remains present at Garden Patch, for example. Significantly, marine turtle meat consumption is still popular in some of the study regions (e.g. [43]) and this analysis indicates that dietary use of turtle meat originated during indigenous times and has endured to the present day despite many cultural/political changes and severe turtle population declines.

The positive species-level taxonomic identification of juvenile turtles at Garden Patch could not have been accomplished without collagen fingerprinting technology because of their less diagnostic morphology when very young combined with their highly fragmented state of preservation. At this site, green turtle remains are found in both midden and mound contexts; however, Kemp's ridley specimens are only found in mound contexts. Since mound deposits are associated with feasting refuse, additional analysis of marine turtle remains may confirm that Kemp's ridley turtles were reserved for non-mundane, supra-household consumption. Furthermore, both green and Kemp's ridley turtles have distinct behaviours and occupy different habitats, suggesting that the inhabitants of Garden Patch were attuned to such differences and possibly practised different capture techniques.

The high success rate of collagen fingerprinting on marine turtle remains from the Virgin Islands, which were stored for almost a hundred years in less-than-optimal microclimatic conditions, opens up a new avenue for reinterpreting well-excavated and documented legacy collections such as this. Our analyses here provide support for the use of this method on other faunal remains preliminarily identified from this collection, including fish, birds and terrestrial mammals. In the absence of morphological expertise, in the case of these Virgin Island sites, biomolecular analyses can build the foundation required before multidisciplinary reanalyses of extended collections can be attempted, which when achieved will have a wide range of implications for the archaeological interpretation of legacy collections and excavations in the future.

The samples selected for analysis in this case study were done so as to test preliminarily the method of collagen fingerprinting on archaeological marine turtle bone, with this being the first analysis of its type. This study included specimens that had been identified morphologically to species level as well as those that were unable to be identified as such owing to heavy fragmentation, juvenile stages of development, lack of expertise in specimen morphological identification or a combination of these factors. The taxonomic resolution that marine turtles display in their collagen signatures has allowed us to confirm, clarify and correct such morphological identifications supporting our case that morphological and biomolecular forms of marine turtle bone identification are complementary to one another in the achievement of species-level identification. Although samples were not selected with specific archaeological or cultural historical hypotheses in mind, the results carry great archaeological significance, including highlighting the potential extinction of a species and the utility of the method where aDNA is not preserved—all of which greatly validate the methodology. The scope for further research of this nature is now vast, including but not limited to (i) exploring changes in target species through time by studying samples from different stratigraphic layers, (ii) investigating which species were targeted for tool making by analysing modified versus unmodified bones, (iii) exploring potential changes in marine turtle distribution through time by analysing greater sample sizes from across different sites and ecological zones.

## 4. Conclusion

Collagen fingerprinting is a valuable resource for species identification in ancient and non-diagnostic bone. Here, we show its capacity to distinguish between all six extant marine turtle genera (*Chelonia*, *Caretta*, *Natator*, *Eretmochelys*, *Lepidochelys* and *Dermochelys*), five of which are known to inhabit the Caribbean and Gulf of Mexico regions (all but *N. depressus*). The unique set of collagen (I) biomarkers developed through this study has allowed taxonomic identification of all collagen-containing archaeological marine turtle bones, and six other Testudines of up to approximately 2500 years old, identified morphologically as 'Cheloniidae', from seven sites across the Caribbean and Florida's Gulf Coast. Identifications by collagen fingerprinting have challenged, refined and complemented previous identifications based on morphological analyses, and have been shown to be superior to aDNA analyses, which were unsuccessful owing to molecular degradation in these tropical environments. We also further support the use of proteomics-derived collagen sequence data for phylogenetic reconstruction, demonstrating trees that are identical in their topology to other molecularly derived trees. Our study provides support that molecular analyses of archaeological marine turtles can allow researchers to link site locales, pre-Columbian time periods and harvesting preferences for particular

species across space and through time, as well as pursuing more taxon-specific questions regarding past marine turtle biogeography, species presence and distribution. The success of this case study indicates that further analyses could assist in reworking historical baseline data for marine turtle distribution across the Caribbean and Gulf, as well as helping determine human exploitation patterns of marine turtles through time. In addition to archaeological questions, the application of collagen fingerprinting on marine turtle bones can be applied to research in palaeontology, biological conservation and international programmes of marine turtle management and protection.

Ethics. This article does not present research with ethical considerations. All handling, sampling and analysis of the modern museum and archaeological marine turtle specimens was conducted in accordance with CITES requirements. No live specimens were included in the analysis.

Data accessibility. All proteome sequence files (.mgf) are available from the Dryad Digital Repository: https://doi.org/10.5061/dryad.6fb6676 [62]. All peptide mass spectra of interest, including both collagen fingerprints and tandem/sequence spectra, are present in the electronic supplementary material for this manuscript.

Authors' contributions. V.L.H. and M.B. conceived and designed the initial experimental plan; M.J.L., S.D.d. and C.T. managed archaeological collections; M.J.L., S.D.d., A.C.K., V.L.H. and M.B. managed the modern collections; M.J.L. managed the morphological identifications of archaeological specimens from Grand Bay and Middle Caicos; V.L.H. performed the collagen fingerprinting experiments, analysed the proteomic data and designed the figures; K.D. and V.L.H. performed aDNA analyses; V.L.H. wrote the manuscript with contributions from all authors. All authors gave final approval for publication.

Competing interests. The authors declare no competing interests.

Funding. We profusely thank The University of Manchester Dean's Award for scholarship funding to V.L.H. as well as support from the Royal Society for fellowship funding (grant no. UF120473) to M.B.

Acknowledgements. We extend our sincere gratitude to the Department of Modern History and World Cultures at the NMD for access to the archaeological marine turtle specimens from O.1 Magens Bay (St Thomas) and O.5 Coral Bay (St John), part of the Gudmund Hatt archaeological collection in Copenhagen. We also wish to gratefully acknowledge the Florida Museum of Natural History and the University of Florida for access to modern samples, with great thanks, in particular, to Dave Blackburn, Coleman Sheehy and Gladys Galdamez. Many thanks also go to the Western Australia Museum Herpetology Collection for a sample of *N. depressus*. We extend our gratitude to Neill Wallis, Scott Fitzpatrick and William Keegan, who provided archaeological samples from Garden Patch, Grand Bay and MC-6, respectively. David Steadman provided helpful comments on an early draft of the manuscript. Many thanks also go to the Biological Mass Spectrometry Core Research Facility (The University of Manchester) for access to their facilities.

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
