## [Reviewer comments · Royal Society Open Science]

Review History

RSOS-191137.R0 (Original submission)

Review form: Reviewer 1

Is the manuscript scientifically sound in its present form?

Yes

Are the interpretations and conclusions justified by the results?

Yes

Is the language acceptable?

Yes

Do you have any ethical concerns with this paper?

No

Have you any concerns about statistical analyses in this paper?

No

Recommendation?

Accept with minor revision (please list in comments)

Comments to the Author(s)

Harvey et al. present an interesting study of collagen sequences derived from marine turtles. I have no major concerns with the manuscript, but do suggest some minor changes and additional explanations to make the research fully reproducible and technically sound.

-Ancient DNA. In several places, the authors refer to ancient DNA research conducted by them on a selected set of specimens. However, no information is provided in the Methods section on this analysis. Please elaborate on the laboratory conditions, extraction methods, bioinformatics, sequence authentication, etc., used for ancient DNA analysis conducted. In addition, please expand on the selection criteria for the few archaeological specimens analyzed for DNA. I see that some of this information is in the supplements, but it needs to be in the main text Methods.

-LC-MS/MS data analysis. Please elaborate on the essential details regarding collagen type I sequence reconstruction during bioinformatic analysis. Particularly, expand on PTM settings and error tolerance settings used, criteria of amino acid polymorphism acceptance (especially when related to novel amino acid substitutions). Finally, explain how you dealt with isobaric residues, either due amino acid substitutions (for example, leucine to isoleucine differentiation), or PTM mass changes (for example, deamidation or oxidations).

-Lines 143-151. This paragraph contains several incorrect statements. For example, recent research available in the literature demonstrates that ancient proteins other than collagen type I survive longer than both ancient DNA and collagen type I. This literature requires citation:
<https://www.biorxiv.org/content/10.1101/407692v1.abstract>
<https://cdn.elifesciences.org/articles/17092/elifesciences-17092-v1.pdf>

-Lines 323-325. Please specify whether the morphologically identified *Caretta caretta* were also studied using collagen fingerprinting. Be specific and provide the sample identifiers.

-Figure 1: Please provide the common names in the figure, as you use the common names and not the Latin names in the associated text. Without the common names, the figure is not immediately relatable to the text.

-Figure 4: Based on the placement it is unclear which archaeological site (of 3) the pie-chart above the island "St. John" relates to. Please be more specific.

Review form: Reviewer 2 (Canan Cakirlar)**Is the manuscript scientifically sound in its present form?**

No

Are the interpretations and conclusions justified by the results?

No

Is the language acceptable?

Yes

Do you have any ethical concerns with this paper?

Yes

Have you any concerns about statistical analyses in this paper?

No

Recommendation?

Major revision is needed (please make suggestions in comments)

Comments to the Author(s)

Dear authors,

I read this paper with great interest. I think it is wonderful that a set of collagen biomarkers for different species of extant sea turtles now exists. One of the weak points of this part of the paper is the limited sample size. Apart from *C. mydas*, all species are represented by one individual only. Increasing the sample size would improve the study. A larger sample size would also support your claim that you found a new, extinct species. The second issue I had with the study is that there is no methodology behind the morphological identifications of the Testudines used in the study (both archaeological and museum). Even if there is, there is no mention of it. This will lead uninformed readers to assume that ZooMS is correcting state-of-the art morphological identifications. Finally the explanations of poor (expected) collagen preservation, archaeological interpretation of the data, and the addition of freshwater turtles are not supported.

Review form: Reviewer 3 (Ren Hirayama)

Is the manuscript scientifically sound in its present form?

Yes

Are the interpretations and conclusions justified by the results?

Yes

Is the language acceptable?

Yes

Do you have any ethical concerns with this paper?

No

Have you any concerns about statistical analyses in this paper?

No

Recommendation?

Accept as is

Comments to the Author(s)

This ms. seems very wonderful for taxonomic identification of marine turtles remains from archeological sites. As many marine turtle remains are disarticulated and very fragmentary, making so serious to identify them as species level. Thus, establishment of this species identification based on preserved collagen will be quite useful for archeological sciences. This should be published very soon!

Decision letter (RSOS-191137.R0)

03-Sep-2019

Dear Dr Buckley

On behalf of the Editors, I am pleased to inform you that your Manuscript RSOS-191137 entitled "Preserved collagen reveals species identity in archaeological marine turtle bones from Caribbean and Florida sites" has been accepted for publication in Royal Society Open Science subject to minor revision in accordance with the referee suggestions. Please find the referees' comments at the end of this email.

The reviewers and handling editors have recommended publication, but also suggest some minor revisions to your manuscript. Therefore, I invite you to respond to the comments and revise your manuscript.

- Ethics statement

- Data accessibility

If you wish to submit your supporting data or code to Dryad (<http://datadryad.org/>), or modify your current submission to dryad, please use the following link:
<http://datadryad.org/submit?journalID=RSOS&manu=RSOS-191137>

- Competing interests

- Authors' contributions

- Acknowledgements

- Funding statement

Because the schedule for publication is very tight, it is a condition of publication that you submit the revised version of your manuscript before 12-Sep-2019. Please note that the revision deadline will expire at 00.00am on this date. If you do not think you will be able to meet this date please let me know immediately.

- 1) A text file of the manuscript (tex, txt, rtf, docx or doc), references, tables (including captions) and figure captions. Do not upload a PDF as your "Main Document";
- 2) A separate electronic file of each figure (EPS or print-quality PDF preferred (either format should be produced directly from original creation package), or original software format);
- 3) Included a 100 word media summary of your paper when requested at submission. Please ensure you have entered correct contact details (email, institution and telephone) in your user account;
- 4) Included the raw data to support the claims made in your paper. You can either include your data as electronic supplementary material or upload to a repository and include the relevant doi

within your manuscript. Make sure it is clear in your data accessibility statement how the data can be accessed;

5) All supplementary materials accompanying an accepted article will be treated as in their final form. Note that the Royal Society will neither edit nor typeset supplementary material and it will be hosted as provided. Please ensure that the supplementary material includes the paper details where possible (authors, article title, journal name).

If your manuscript is newly submitted and subsequently accepted for publication, you will be asked to pay the article processing charge, unless you request a waiver and this is approved by Royal Society Publishing. You can find out more about the charges at <http://rsos.royalsocietypublishing.org/page/charges>. Should you have any queries, please contact opscience@royalsociety.org.

Kind regards,
Andrew Dunn
Royal Society Open Science Editorial Office
Royal Society Open Science
opscience@royalsociety.org

on behalf of Professor Matthew Collins (Associate Editor) and Kevin Padian (Subject Editor)
opscience@royalsociety.org

Associate Editor Comments to Author (Professor Matthew Collins):

Associate Editor: 1

Comments to the Author:

Dear Mike

A mixed bag of reviewers with a full range of recommendations.

In my view, the MS actually only needs minor revision, and I have suggested some course of actions

The first reviewer is positive with a few minor suggestions for improvements. The second suggest improvements to the paper, involving acknowledging work on the osteology and osteometry of *Chelonia* and the issues of identification of juvenile and small specimens; these

seem sensible suggestions to strengthen the paper, in the light of the focus of the work. The referee also asks you to acknowledge more fully in the text the small sample size of some of your examples, I leave to make your own judgement call on this.

The final Reviewer, well the final reviewer.....

Some minor typos and suggestions

Line 147 indispensable -> indispensable

Line 163 Paragraph tense, both is and was are both used. Check your intended tenses

Line 182 of the archaeological sites, 'the' is not essential here.

Line 250 spelling of co-crystallised

Line 376 southern-most to southernmost

Line 436 sea grass -> seagrass

Reviewer 1

Harvey et al. present an interesting study of collagen sequences derived from marine turtles. I have no major concerns with the manuscript, but do suggest some minor changes and additional explanations to make the research fully reproducible and technically sound.

-Ancient DNA. In several places, the authors refer to ancient DNA research conducted by them on a selected set of specimens. However, no information is provided in the Methods section on this analysis.

Please elaborate on the laboratory conditions, extraction methods, bioinformatics, sequence authentication, etc., used for ancient DNA analysis conducted. In addition, please expand on the selection criteria for the few archaeological specimens analyzed for DNA. I see that some of this information is in the supplements, but it needs to be in the main text Methods.

Assoc Editor - I appreciate the reviewer's point, and while I agree that you should elaborate the selection criteria, I leave it to your decision as to whether you bring the aDNA methods into the main body of the text.

-LC-MS/MS data analysis. Please elaborate on the essential details regarding collagen type I sequence reconstruction during bioinformatic analysis. Particularly, expand on PTM settings and error tolerance settings used, criteria of amino acid polymorphism acceptance (especially when related to novel amino acid substitutions). Finally, explain how you dealt with isobaric residues, either due amino acid substitutions (for example, leucine to isoleucine differentiation), or PTM mass changes (for example, deamidation or oxidations).

-Lines 143-151. This paragraph contains several incorrect statements. For example, recent research available in the literature demonstrates that ancient proteins other than collagen type I survive longer than both ancient DNA and collagen type I. This literature requires citation:

<https://www.biorxiv.org/content/10.1101/407692v1.abstract>

<https://cdn.elifesciences.org/articles/17092/elifesciences-17092-v1.pdf>

-Lines 323-325. Please specify whether the morphologically identified *Caretta caretta* were also studied using collagen fingerprinting. Be specific and provide the sample identifiers.

-Figure 1: Please provide the common names in the figure, as you use the common names and not

the Latin names in the associated text. Without the common names, the figure is not immediately relatable to the text.

-Figure 4: Based on the placement it is unclear which archaeological site (of 3) the pie-chart above the island "St. John" relates to. Please be more specific.

Assoc Editor - A line connecting the piechart to the site will suffice

Reviewer 2

I would not recommend publishing this study in its current format for three reasons. But the main reason is that the osteomorphological identifications have no methodology described (that's half the study), and the authors claim to correct the osteological identifications with ZooMS (the other half of the study).

1. The manuscript depicts a false (and not useful) dichotomy between morphology and ZooMS identifications. The picture depicted here is simply created by the inadequacy of the morphological research and does not reflect the status of research in this field.
2. The suggestion of an extinct marine turtle species in the Atlantic is based on very small sample size.
3. There is no archaeological time-period resolution.

This paper explains a first attempt to read the collagen fingerprint of extant sea turtles. It is an important contribution towards the species identification of cryptic sea turtle remains. As the author's highlight, species identification in turtles is essential to understand their past distributions and interactions with humans. One of the premises of the paper is that ZooMS corrects morphological identifications.

First, the authors' exciting claim that ZooMS can correct morphological identification and complements is potentially true. However, this paper does not test or confirm this claim -yet. This is because there is virtually not a word in the paper on how the researchers identified the marine turtles from their skeletal remains (recent, museum, or archaeological). Comparative testudine osteomorphology is a research field which is essentially ignored in the paper.

Second, authors write (191-195) that morphological similarities especially among juvenile individuals hamper identification down to species level. However, they provide no citation. There is a good amount of literature on the comparative morphology of extinct and extant turtles. Some of this literature has recently be reviewed and published. Recent studies on the skeletons of recent specimens have shown that there are many more landmarks on the appendicular skeletons of Cheloniidae that allow identification down to species level even in fragmented archaeological specimens. Unlike mammals, turtle bones grow incrementally. Like fish, their allometry is simpler, and growth does not involve epiphyseal fusion. Unlike mammals, juvenile individuals display skeletal morphologies rather similar to adult specimens.

In the Abstract, Introduction, and Conclusions, authors highlight how ZooMS correctly identifies *C. caretta* identifications based on morphology, but in Lines 320-325 and in S1, it is shown that the identifications were, in fact, tentative (for the wrong reasons probably, but still -they were 'cf'). It is also quite possible (as I gather from lines 330-336) that the 'juveniles' in one of the archaeological sites (Garden Patch) were identified as juveniles because they actually belong to the freshwater turtle species that never attain sizes as large as marine turtles.

Second, the authors suggest that collagen preservation in the tropics is unexpected. However, they do not offer any support for this suggestion. Which specific factors in the tropics may have contributed to for ZooMS favourable preservation conditions?

Lines 383-388 discuss a very vague finding. It is good to know the success rate of collagen extraction. But again: How were the specimens identified to species?

Third, the claims for a different, yet extinct species of *Chelonia* (Lines 358-365), I find, are not mature, because they are based on rather thin evidence (small sample size from restricted geography).

Finally, the section on the archaeological interpretation is interesting. However, it covers more than 2000 years, but the narrative lacks comments on diachronic trends. Table 1 in Supp M. does not provide phasings or datings per specimen either, reducing its value to reconstruct an archaeological narrative of what may have happened, how turtle species were distributed in which time periods, and of course why.

I read this paper with great interest. I think it is wonderful that a set of collagen biomarkers for different species of extant sea turtles now exists. One of the weak points of this part of the paper is the limited sample size. Apart from *C. mydas*, all species are represented by one individual only. Increasing the sample size would improve the study. A larger sample size would also support your claim that you found a new, extinct species. The second issue I had with the study is that there is no methodology behind the morphological identifications of the Testudines used in the study (both archaeological and museum). Even if there is, there is no mention of it. This will lead uninformed readers to assume that ZooMS is correcting state-of-the art morphological identifications. Finally, the explanations of poor (expected) collagen preservation, archaeological interpretation of the data, and the addition of freshwater turtles are not supported.

This paper explains a first attempt to read the collagen fingerprint of extant sea turtles. It is an important contribution towards the species identification of cryptic sea turtle remains. As the author's highlight, species identification in turtles is essential to understand their past distributions and interactions with humans. One of the premises of the paper is that ZooMS corrects morphological identifications.

First, the authors' exciting claim that ZooMS can correct morphological identification and complements is potentially true. However, this paper does not test or confirm this claim -yet. This is because there is virtually not a word in the paper on how the researchers identified the marine turtles from their skeletal remains (recent, museum, or archaeological). The comparative testudine osteomorphology is a research field.

Second, authors write (191-195) that morphological similarities, especially among juvenile individuals, hamper identification down to species level. However, they provide no citation. There is a good amount of literature on the comparative morphology of extinct and extant turtles. Some of this literature has recently been reviewed and published. Recent studies on the skeletons of recent specimens have shown that there are many more landmarks on the appendicular skeletons of Cheloniidae that allow identification down to species level even in fragmented archaeological specimens. Unlike mammals, turtle bones grow incrementally. Like fish, their allometry is simpler, and growth does not involve epiphyseal fusion. Unlike mammals, juvenile individuals display skeletal morphologies rather similar to adult specimens.

In the Abstract, Introduction, and Conclusions, authors highlight how ZooMS correct the *C. caretta* identifications based on morphology, but in Lines 320-325 and in S1, it is shown that the

identifications were in fact tentative (for the wrong reasons probably, but still -they were 'cf'). It is also quite possible (as I gather from lines 330-336) that the 'juveniles' in one of the archaeological sites (Garden Patch) were identified as juveniles because they actually belong to the freshwater turtle species that never attain sizes as large as marine turtles.

Second, the authors suggest that collagen preservation in the tropics is unexpected. However, they do not offer any support for this suggestion. Which specific factors in the tropics may have contributed to for ZooMS favourable preservation conditions?

Lines 383-388 discuss a very vague finding. It is good to know the success rate of collagen extraction. But again: How were the specimens identified to species?

Third, the claims for a different, yet extinct species of *Chelonia* (Lines 358-365), I find, are not mature, because they are based on rather thin evidence (small sample size from restricted geography).

Finally, the section on the archaeological interpretation is interesting. However, it covers more than 2000 years, but the narrative lacks comments on diachronic trends. Table 1 in Supp M. does not provide phasings or datings per specimen either, reducing its value to reconstruct an archaeological narrative of what may have happened, how turtle species were distributed in which time periods, and of course why.

Length

The paper is a little bit too long. Information on the archaeological sites can be provided in the Sup.M., rather than the main body text.

Moreover, it is not clear how the analyses of non-marine turtles help answer the research question. The research question, as framed in the Introduction, focuses on distinguishing marine turtle species from their skeletal remains.

However, I would be curious to read more about how collagen evolves in turtles, whether and how the evolutionary rate and pathways of turtle proteins make ZooMS different in these species than other taxa. This issue is barely touched (lines 301-304).

Statistics

Authors have sampled three *C. mydas* individuals, and one individual per taxon for the remaining six extant marine turtle species. Table 1 and the SM indicate that the differences in the collagen biomarkers between *C. caretta* and *C. mydas* are not remarkable. If the differences are significant, these should be demonstrated. Whether this can be achieved using the number of modern samples analysed is, to me, questionable.

Reviewer 3

This ms. seems very wonderful for taxonomic identification of marine turtles remains from archeological sites. As many marine turtle remains are disarticulated and very fragmentary, making so serious to identify them as species level. Thus, establishment of this species identification based on preserved collagen will be quite useful for archeological sciences. This should be published very soon!

Associate Editor: 2

Comments to the Author:
(There are no comments.)

Reviewer comments to Author:

Reviewer: 1

Comments to the Author(s)

Harvey et al. present an interesting study of collagen sequences derived from marine turtles. I have no major concerns with the manuscript, but do suggest some minor changes and additional explanations to make the research fully reproducible and technically sound.

-Ancient DNA. In several places, the authors refer to ancient DNA research conducted by them on a selected set of specimens. However, no information is provided in the Methods section on this analysis. Please elaborate on the laboratory conditions, extraction methods, bioinformatics, sequence authentication, etc., used for ancient DNA analysis conducted. In addition, please expand on the selection criteria for the few archaeological specimens analyzed for DNA. I see that some of this information is in the supplements, but it needs to be in the main text Methods.

-LC-MS/MS data analysis. Please elaborate on the essential details regarding collagen type I sequence reconstruction during bioinformatic analysis. Particularly, expand on PTM settings and error tolerance settings used, criteria of amino acid polymorphism acceptance (especially when related to novel amino acid substitutions). Finally, explain how you dealt with isobaric residues, either due amino acid substitutions (for example, leucine to isoleucine differentiation), or PTM mass changes (for example, deamidation or oxidations).

-Lines 143-151. This paragraph contains several incorrect statements. For example, recent research available in the literature demonstrates that ancient proteins other than collagen type I survive longer than both ancient DNA and collagen type I. This literature requires citation:
<https://www.biorxiv.org/content/10.1101/407692v1.abstract>
<https://cdn.elifesciences.org/articles/17092/elif-17092-v1.pdf>

-Lines 323-325. Please specify whether the morphologically identified *Caretta caretta* were also studied using collagen fingerprinting. Be specific and provide the sample identifiers.

-Figure 1: Please provide the common names in the figure, as you use the common names and not the Latin names in the associated text. Without the common names, the figure is not immediately relatable to the text.

-Figure 4: Based on the placement it is unclear which archaeological site (of 3) the pie-chart above the island "St. John" relates to. Please be more specific.

Reviewer: 2

Comments to the Author(s)

Dear authors,

I read this paper with great interest. I think it is wonderful that a set of collagen biomarkers for different species of extant sea turtles now exists. One of the weak points of this part of the paper is the limited species sample size. Apart from *C. mydas*, all species are represented by one individual only. Increasing the sample size would improve the study. A larger sample size would also support your claim that you found a new, extinct species. The second issue I had with the study is that there is no methodology behind the morphological identifications of the Testudines used in the study (both archaeological and museum). Even if there is, there is no mention of it. This

will lead uninformed readers to assume that ZooMS is correcting state-of-the art morphological identifications. Finally the explanations of poor (expected) collagen preservation, archaeological interpretation of the data, and the addition of freshwater turtles are not supported.

Reviewer: 3

Comments to the Author(s)

This ms. seems very wonderful for taxonomic identification of marine turtles remains from archeological sites. As many marine turtle remains are disarticulated and very fragmentary, making so serious to identify them as species level. Thus, establishment of this species identification based on preserved collagen will be quite useful for archeological sciences. This should be published very soon!

Author's Response to Decision Letter for (RSOS-191137.R0)

See Appendix A.

Decision letter (RSOS-191137.R1)

04-Oct-2019

Dear Dr Buckley,

I am pleased to inform you that your manuscript entitled "Preserved collagen reveals species identity in archaeological marine turtle bones from Caribbean and Florida sites" is now accepted for publication in Royal Society Open Science.

Additionally, we note that the e-mail address "mlefebvre@ufl.edu" is not currently receiving our messages. Please can you check this and/or provide an alternative e-mail address from your colleague?

Kind regards,
Anita Kristiansen
Royal Society Open Science Editorial Office
Royal Society Open Science
openscience@royalsociety.org

on behalf of Professor Matthew Collins (Associate Editor) and Kevin Padian (Subject Editor)
openscience@royalsociety.org

Follow Royal Society Publishing on Twitter: [@RSocPublishing](https://twitter.com/RSocPublishing)
Follow Royal Society Publishing on Facebook:
<https://www.facebook.com/RoyalSocietyPublishing.FanPage/>
Read Royal Society Publishing's blog: <https://blogs.royalsociety.org/publishing/>

Appendix A

Response to Referees

RE: Preserved collagen reveals species identity in archaeological marine turtle bones from Caribbean and Florida sites

We gratefully acknowledge the constructive comments from all the reviewers of our manuscript. Our responses are shown below.

Associate Editor Comments to Author (Professor Matthew Collins):

Associate Editor: 1

Comments to the Author:

Some minor typos and suggestions

Line 147 indispensable -> indispensable

We have amended this typo

Line 163 Paragraph tense, both is and was are both used. Check your intended tenses

We have changed to past tense throughout this paragraph

Line 182 of the archaeological sites, 'the' is not essential here.

We have removed this definite article as suggested

Line 250 spelling of co-crystalised

We have now corrected this

Line 376 southern-most to southernmost

We have now corrected this

Line 436 sea grass -> seagrass

We have now corrected this

Reviewer 1

Harvey et al. present an interesting study of collagen sequences derived from marine turtles. I have no major concerns with the manuscript, but do suggest some minor changes and additional explanations

to make the research fully reproducible and technically sound.

-Ancient DNA. In several places, the authors refer to ancient DNA research conducted by them on a selected set of specimens. However, no information is provided in the Methods section on this analysis.

Please elaborate on the laboratory conditions, extraction methods, bioinformatics, sequence authentication, etc., used for ancient DNA analysis conducted. In addition, please expand on the selection criteria for the few archaeological specimens analyzed for DNA. I see that some of this information is in the supplements, but it needs to be in the main text Methods.

Assoc Editor - I appreciate the reviewer's point, and while I agree that you should elaborate the selection criteria, I leave it to your decision as to whether you bring the aDNA methods into the main body of the text.

Whilst we agree that the ancient DNA methodology would be best placed in the main body of the text, we are unfortunately restricted by the word count that this section requires to be explained sufficiently. As such, we have kept it in the supplementary information but we have updated it with additional information regarding the methods and procedure, as per the reviewer's suggestion:

"Samples were set for sequencing to Eurofins Genomics and sequences were analysed on Geneious Prime. Marine turtle reference genomes were downloaded from NCBI (www.ncbi.nlm.nih.gov) and default algorithms for alignment were used."

-LC-MS/MS data analysis. Please elaborate on the essential details regarding collagen type I sequence reconstruction during bioinformatic analysis. Particularly, expand on PTM settings and error tolerance settings used, criteria of amino acid polymorphism acceptance (especially when related to novel amino acid substitutions). Finally, explain how you dealt with isobaric residues, either due amino acid substitutions (for example, leucine to isoleucine differentiation), or PTM mass changes (for example, deamidation or oxidations).

We have added the following to the relevant section in the Materials and Methods:

"ET searches used the following criteria: peptide tolerance of ± 5 ppm, MS/MS fragment ion mass value tolerance of 0.5 Da, fixed carbamidomethyl modification of cysteine (mass shift = +57.02 Da), variable modifications for the oxidation of lysine (K) and proline (P) (mass shift = +15.99 Da), and the allowance of one missed cleavage. A final decoy search was used to mine the finished sequences, filtered to only include tryptic peptides that scored higher than the highest false positive result with all other positions replaced with an 'X'. Applied decoy search criteria was as above except with two missed cleavages, additional variable deamidation of asparagine (N) and glutamine (Q) modifications (mass shift = +0.98 Da) and variable oxidation of methionine (M). Fragment ion spectra for novel amino acid substitution biomarkers were manually examined for quality before selection (Supplementary Figure S2). Where the isobaric residues leucine (L) and isoleucine (I), provided ambiguity, sequences were completed with the residue L to provide consistency. We found no ambiguity between lysine (K) and glutamine (Q) residues as these sites are highly conserved in

marine turtles, with the former also providing a tryptic digest site. Where the post-translational modification of hydroxylation provided ambiguity with an alanine (A) to serine (S) substitution (mass shift of either = +15.99 Da) the sequence was completed with the residue congruent with the published sequence for *C. mydas* (BLAST/P).”

-Lines 143-151. This paragraph contains several incorrect statements. For example, recent research available in the literature demonstrates that ancient proteins other than collagen type I survive longer than both ancient DNA and collagen type I. This literature requires citation:

<https://www.biorxiv.org/content/10.1101/407692v1.abstract>
<https://cdn.elifesciences.org/articles/17092/elif-17092-v1.pdf>

These are very interesting papers that show the longevity of other proteins. We have added these two citations and have adapted the sentence as follows to allow for this information:

“Collagen type 1 (“collagen (I)”), a fibrous protein found ubiquitously in vertebrate bone, has been shown to survive longer post-deposition than many other informative biomolecules, including ancient DNA (aDNA) (e.g. (26), although see (27, 28).”

-Lines 323-325. Please specify whether the morphologically identified *Caretta caretta* were also studied using collagen fingerprinting. Be specific and provide the sample identifiers.

We have changed this sentence to read:

“However, the absence of the loggerhead turtle, *Caretta caretta*, was not specifically predicted, particularly as a number of samples (n=12; Garden Patch samples 1, 2, 8–17) were morphologically identified, albeit cautiously, as *Caretta caretta* prior to collagen analysis (Supplementary Table S1).”

-Figure 1: Please provide the common names in the figure, as you use the common names and not the Latin names in the associated text. Without the common names, the figure is not immediately relatable to the text.

We have changed Figure 1 to use common names rather than Latin names and we have tweaked the caption and the text to suit.

-Figure 4: Based on the placement it is unclear which archaeological site (of 3) the pie-chart above the island "St. John" relates to. Please be more specific.

We have changed the caption to Figure 4 to state that that pie charts relate to the whole island in each case. This should remove the ambiguity. We have also shifted the position of the pie chart that relates to St. John and placed it more centrally.

Assoc Editor - A line connecting the piechart to the site will suffice

We played around with adding connecting lines but the addition of these looked messy so we have opted for the above.

Reviewer 2

I would not recommend publishing this study in its current format for three reasons. But the main reason is that the osteomorphological identifications have no methodology described (that's half the study), and the authors claim to correct the osteological identifications with ZooMS (the other half of the study).

See below.

1. The manuscript depicts a false (and not useful) dichotomy between morphology and ZooMS identifications. The picture depicted here is simply created by the inadequacy of the morphological research and does not reflect the status of research in this field.

We respectfully disagree that our claims regarding the identification of archaeological sea turtle specimens “is simply created by the inadequacy of the morphological research and does not reflect the status of research in this field”. We acknowledge that there have been great strides made in the identification of sea turtle morphology overall – including reference guides, etc. We were remiss to not acknowledge recent morphological work addressing these challenges in archaeological settings. We have edited the paper to reflect this and now cite the most recent reference regarding this matter.

However, as demonstrated in our study we are speaking to zooarchaeological identification specifically and the challenges often particular to archaeological contexts and taphonomic processes affecting the state and preservation of bone. Across many archaeological sites with sea turtle remains, the remains are often highly fragmentary (i.e. not complete elements, or lacking morphological characteristics of specific species on broken remains), eroded, and/or completely disassociated from original contexts of sea turtle capture and butchering. In our study areas across Florida and the Caribbean we recover hundreds, if not thousands, of broken, highly fragmentary, and eroded sea turtle specimens from midden contexts containing thousands of other types of animal remains from various taxa. The zooarchaeological identification of sea turtle specimens to level of species is indeed a real disciplinary challenge within zooarchaeology with significant impacts on interpretations (see Reviewer 3’s comment). What our study accurately reflects is the reality of sea turtle zooarchaeological identification; where in the case of high bone fragmentation and mixed contexts of deposition (e.g., middens), it is very difficult to confidently achieve species-level identifications.

We do not believe we have depicted a dichotomy (false or otherwise) between morphology and ZooMS. Rather, as the other reviewers appreciated, we are showing how zooarchaeological morphological analysis and ZooMS are complementary methods for advancing zooarchaeological methods of specimen identification and as taxonomically accurate as possible species data.

We have made edits to the paper to clarify these issues and similar points made by the other reviewers.

2. The suggestion of an extinct marine turtle species in the Atlantic is based on very small sample size.

Whilst we acknowledge that it is a small sample size (4 from a total of 130 marine turtle bone samples), the evidence shown by the proteomic data indicates this as an undescribed species of marine turtle, most closely related to the green turtle (*Chelonia mydas*). The small sample size does not affect this claim, which is founded in the protein data we present, as we have taken into account every extant species that has been described. Assuming that it is not a modification, the only alternative is that it could be an undescribed extant species, but this would be equally significant. Whilst we haven't yet been able to fully verify this claim using DNA analysis, we have MS data that support it. However, because we cannot rule out diagenetic factors we tone down the language to be more speculative towards the potential identification of extinct/undescribed taxon. We others, or ourselves, are able to carry out further screening of archaeological marine turtle bones at these sites and beyond using the methodologies we present here to increase the sample size and confidence in the proposed results.

3. There is no archaeological time-period resolution.

We have edited the paper to clarify that the archaeological sites represented span pre-Columbian (pre-AD 1492) and Historic (post AD 1492) periods. Also, we state in the Introduction that our sea turtle samples lived between 500-2500 years ago. The time period(s) of each site are included in the site and assemblage descriptions.

This paper explains a first attempt to read the collagen fingerprint of extant sea turtles. It is an important contribution towards the species identification of cryptic sea turtle remains. As the author's highlight, species identification in turtles is essential to understand their past distributions and interactions with humans. One of the premises of the paper is that ZooMS corrects morphological identifications.

Thank you for your positive comments here. Please note however that ZooMS correcting morphological identification is not one of the premises of this paper, as it has been shown in numerous other papers.

First, the authors' exciting claim that ZooMS can correct morphological identification and complements is potentially true. However, this paper does not test or confirm this claim -yet. This is because there is virtually not a word in the paper on how the researchers identified the marine turtles from their skeletal remains (recent, museum, or archaeological). Comparative testudine osteomorphology is a research field which is essentially ignored in the paper.

We have edited the paper to clarify how the archaeological specimens were identified prior to the ZooMS analysis. All Garden Patch, MC-6 and Grand Bay samples were morphologically analysed using complete, disarticulated, modern sea turtle skeletons as a basis for comparison. All St. John and St. Thomas samples were not systematically analysed morphologically prior to this study due to the absence of morphological expertise and whilst this information was included initially, we have elaborated on it in the manuscript. These details also enhance our stance that collagen fingerprinting and morphological analyses are complementary in the identification of archaeological bone material.

Second, authors write (191-195) that morphological similarities, especially among juvenile individuals, hamper identification down to species level. However, they provide no citation. There is a good amount of literature on the comparative morphology of extinct and extant turtles. Some of this literature has recently been reviewed and published. Recent studies on the skeletons of recent specimens have shown that there are many more landmarks on the appendicular skeletons of Cheloniidae that allow identification down to species level even in fragmented archaeological specimens. Unlike mammals, turtle bones grow incrementally. Like fish, their allometry is simpler, and growth does not involve epiphyseal fusion. Unlike mammals, juvenile individuals display skeletal morphologies rather similar to adult specimens.

Yes, we agree with the reviewer overall. However, as is the case with many archaeological bone specimens, the sea turtle specimens from the assemblages used in our study are characteristic of the regions of study, i.e. the sea turtle specimens are highly fragmented and broken, surfaces are eroded, and the majority are from carapace or plastron elements – and not cranial or post cranial elements (e.g., vertebrae, limbs, etc.)—we have added this detail into the ‘Sampled archaeological sites’ section. Many studies of sea turtle morphology are based on complete specimens and do indicate markers that can be used to identify species. However, in many cases zooarchaeological sea turtle specimens, especially those from the carapace and plastron, are often so incomplete that one cannot confidently assigned species (there are several published examples of this).

In the Abstract, Introduction, and Conclusions, authors highlight how ZooMS correctly identifies *C. caretta* identifications based on morphology, but in Lines 320-325 and in S1, it is shown that the identifications were, in fact, tentative (for the wrong reasons probably, but still -they were 'cf'). It is also quite possible (as I gather from lines 330-336) that the 'juveniles' in one of the archaeological sites (Garden Patch) were identified as juveniles because they actually belong to the freshwater turtle species that never attain sizes as large as marine turtles.

Yes, the reviewer is correct that the species identifications were in fact tentative (e.g., cf.). This was intentional as the analyst choose to exercise caution knowing that it is difficult to assign species-level identifications to fragmented zooarchaeological remains. We do not think this is a weakness in the paper, but rather an honest depiction of previous morphological analysis. In this case, the ZooMS results did bolster the results of morphological analysis.

We have now noted that the identification were “*cf. Caretta caretta*” in the main body of the manuscript. We have also added in the following sentence:

“The remains of these species are likely to have been mistaken for juvenile marine turtles owing to their smaller size and deposition amongst true juvenile marine turtle species.”

Second, the authors suggest that collagen preservation in the tropics is unexpected. However, they do not offer any support for this suggestion. Which specific factors in the tropics may have contributed to for ZooMS favourable preservation conditions?

We state in our manuscript that collagen preservation in our samples from the tropics was “higher than anticipated given (a) the proposed age of the bones at ≥500–2500 years old, and (b) the tropical

(high temperature, high humidity) region in which the bones were deposited, which does not typically favour molecular preservation (52).”

The citation we include is a nice case study showing how factors prevalent in tropical environments do not typically favour molecular preservation (Pestle WJ, Colvard M. Bone collagen preservation in the tropics: a case study from ancient Puerto Rico. *Journal of Archaeological Science*. 2012;39(7):2079–90).

We do not mean to imply that preservation is unexpected, more emphasise the significance over likelihood of other biomolecules surviving.

Lines 383-388 discuss a very vague finding. It is good to know the success rate of collagen extraction. But again: How were the specimens identified to species?

The paper has been edited to address this concern (also see above).

Third, the claims for a different, yet extinct species of *Chelonia* (Lines 358-365), I find, are not mature, because they are based on rather thin evidence (small sample size from restricted geography).

Please see our response above.

Finally, the section on the archaeological interpretation is interesting. However, it covers more than 2000 years, but the narrative lacks comments on diachronic trends. Table 1 in Supp M. does not provide phasings or datings per specimen either, reducing its value to reconstruct an archaeological narrative of what may have happened, how turtle species were distributed in which time periods, and of course why.

This is true. We do not address diachronic trends in this paper as its focus is primarily methodological and related to taxonomic identification. As explained in the paragraph before the Conclusion, our focus of this study is providing a new method from which we can generate more accurate zooarchaeological identifications to the species level and data for future research. Diachronic trends are certainly important to consider in future research and we point this out specifically as a suggestion for future research.

I read this paper with great interest. I think it is wonderful that a set of collagen biomarkers for different species of extant sea turtles now exists. One of the weak points of this part of the paper is the limited sample size. Apart from *C. mydas*, all species are represented by one individual only. Increasing the sample size would improve the study. A larger sample size would also support your claim that you found a new, extinct species. The second issue I had with the study is that there is no methodology behind the morphological identifications of the Testudines used in the study (both archaeological and museum). Even if there is, there is no mention of it. This will lead uninformed readers to assume that ZooMS is correcting state-of-the-art morphological identifications. Finally, the explanations of poor (expected) collagen preservation, archaeological interpretation of the data, and the addition of freshwater turtles are not supported.

The ancient sample size was 130 samples, which is adequate for the outcomes of the manuscript. This is the first study looking and collagen fingerprinting in marine turtles. As such, this manuscript demonstrates that this application of ZooMS is viable to achieve species-level identification (or genus-level in the case of the ridleys) in archaeological marine turtle bone. The underlying science would not be affected by an increase in archaeological sample size. Moreover, publishing the methods and biomarkers paves the way for the screening of much larger sample sizes than the scope of this study could support.

The modern sample size consisted of one specimen for each of the extant marine turtle species, with the exception of the green turtle where we obtained three specimens to cover both the Atlantic and Pacific variants. Population-level collagen amino acid sequence disparities have never yet been recorded in fingerprints for any species, which means that one confidently-identified modern specimen per species of marine turtle should be sufficient, but further reference material would be sought with future studies on different assemblages where appropriate (whether by ourselves or by others). The addition of the Atlantic and Pacific green turtle variants were so that we could test whether the samples of *Chelonia* sp. were more likely to be from a different population or a different species. Because all our modern samples across the globe had a biomarker peak at m/z 3007 we conclude that this is most likely a new species.

This paper explains a first attempt to read the collagen fingerprint of extant sea turtles. It is an important contribution towards the species identification of cryptic sea turtle remains. As the author's highlight, species identification in turtles is essential to understand their past distributions and interactions with humans. One of the premises of the paper is that ZooMS corrects morphological identifications.

Length

The paper is a little bit too long. Information on the archaeological sites can be provided in the Sup.M., rather than the main body text.

We feel that, given the word count permits, we would like to keep the archaeological site information in the main body of the manuscript rather than the supplementary as it carries interest and relevance to our archaeology readers, as well as context to the article as a whole.

Moreover, it is not clear how the analyses of non-marine turtles help answer the research question. The research question, as framed in the Introduction, focuses on distinguishing marine turtle species from their skeletal remains.

The analysis of non-marine turtles was included to simply identify the samples that were suspected as marine turtles morphologically, but in fact were not.

However, I would be curious to read more about how collagen evolves in turtles, whether and how the evolutionary rate and pathways of turtle proteins make ZooMS different in these species than other taxa. This issue is barely touched (lines 301-304).

We agree that this would be a very interesting study, and one that would require its own dedicated manuscript, not within the scope of this study. As such we consider that our brief discussion of collagen evolution in turtles fitting given the lack of current literature on this subject, where no one has yet studied the evolutionary rates and pathways of turtle collagen evolution in relation to themselves or with other taxa.

Statistics

Authors have sampled three *C. mydas* individuals, and one individual per taxon for the remaining six extant marine turtle species. Table 1 and the SM indicate that the differences in the collagen biomarkers between *C. caretta* and *C. mydas* are not remarkable. If the differences are significant, these should be demonstrated. Whether this can be achieved using the number of modern samples analysed is, to me, questionable.

See also comment above. There are 20 amino acid substitutions between the collagen (I) sequences of *Chelonia mydas* and *Caretta caretta* (we provide the full sequences in Supplementary Table S4 and have now added a small table that helps to visualise these changes). Not all of these tryptic peptides are seen in the collagen fingerprints, which typically only show ~40% of all peptides. Therefore in Table 1 we have simply selected two peptides that do appear in the collagen fingerprints of all of the species (and are less conserved) to act as biomarkers for taxon separation. We believe that this data should not change even if we increased the number of modern specimens for any of the species.

Reviewer 3

This ms. seems very wonderful for taxonomic identification of marine turtles remains from archeological sites. As many marine turtle remains are disarticulated and very fragmentary, making so serious to identify them as species level. Thus, establishment of this species identification based on preserved collagen will be quite useful for archeological sciences. This should be published very soon!

Associate Editor: 2

Comments to the Author:

(There are no comments.)